# Dynamic analysis of circulating tumor DNA to predict the prognosis and monitor the treatment response of patients with metastatic triple-negative breast cancer: A prospective study

Yajing Chi[1,2], Mu Su[3], Dongdong Zhou[1], Fangchao Zheng[1], Baoxuan Zhang[1], Ling Qiang[1], Guohua Ren[1], Lihua Song[1], Bing Bu[1], Shu Fang[1], Bo Yu[3], Jinxing Zhou[3], Jinming Yu[4], Huihui Li[1]*

[1]Department of Breast Medical Oncology, Shandong Cancer Hospital and Institute, Shandong First Medical University and Shandong Academy of Medical Sciences, Jinan, China; [2]School of Medicine, Nankai University, Tianjin, China; [3]Department of Bioinformatics, Berry Oncology Corporation, Beijing, China; [4]Department of Radiation Oncology, Shandong Cancer Hospital and Institute, Shandong First Medical University and Shandong Academy of Medical Sciences, Jinan, China

*For correspondence:
huihuili82@163.com

## Abstract

**Background:** Limited data are available on applying circulating tumor DNA (ctDNA) in metastatic triple-negative breast cancer (mTNBC) patients. Here, we investigated the value of ctDNA for predicting the prognosis and monitoring the treatment response in mTNBC patients.

**Methods:** We prospectively enrolled 70 Chinese patients with mTNBC who had progressed after ≤2 lines of chemotherapy and collected blood samples to extract ctDNA for 457-gene targeted panel sequencing.

**Results:** Patients with ctDNA+, defined by 12 prognosis-relevant mutated genes, had a shorter progression-free survival (PFS) than ctDNA− patients (5.16 months vs. 9.05 months, p=0.001), and ctDNA+ was independently associated with a shorter PFS (HR, 95% CI: 2.67, 1.2–5.96; p=0.016) by multivariable analyses. Patients with a higher mutant-allele tumor heterogeneity (MATH) score (≥6.316) or a higher ctDNA fraction (ctDNA%≥0.05) had a significantly shorter PFS than patients with a lower MATH score (5.67 months vs.11.27 months, p=0.007) and patients with a lower ctDNA% (5.45 months vs. 12.17 months, p<0.001), respectively. Positive correlations with treatment response were observed for MATH score (R=0.24, p=0.014) and ctDNA% (R=0.3, p=0.002), but not the CEA, CA125, or CA153. Moreover, patients who remained ctDNA+ during dynamic monitoring tended to have a shorter PFS than those who did not (3.90 months vs. 6.10 months, p=0.135).

**Conclusions:** ctDNA profiling provides insight into the mutational landscape of mTNBC and may reliably predict the prognosis and treatment response of mTNBC patients.

**Funding:** This work was supported by the National Natural Science Foundation of China (Grant No. 81902713), Natural Science Foundation of Shandong Province (Grant No. ZR2019LZL018), Breast Disease Research Fund of Shandong Provincial Medical Association (Grant No. YXH2020ZX066), the Start-up Fund of Shandong Cancer Hospital (Grant No. 2020-PYB10), Beijing Science and Technology Innovation Fund (Grant No. KC2021-ZZ-0010-1).

## Editor's evaluation

This valuable study advances our understanding of the predictive role of circulating tumor DNA (ctDNA) in the prognosis of patients with mTNBC as well as other malignant tumors. The evidence supporting the conclusions is solid, with rigorous analysis of the association between ctDNA (ctDNA-positive or not) with the progression-free survival (PFS) of patients. The work will be of broad interest to clinicians, medical researchers, and scientists working in breast cancer and cancer detection.

## Introduction

Breast cancer is the most common malignant tumor and the leading cause of cancer-related deaths in women worldwide (*Sung et al., 2021*). Triple-negative breast cancer (TNBC) represents 15–20% of all breast cancer cases and exhibits a more aggressive phenotype (with a poorer prognosis) than non-TNBC (*Foulkes et al., 2010*; *Li et al., 2017*; *Malorni et al., 2012*). Due to the absence of human epidermal growth factor receptor 2 (HER2), estrogen receptor (ER), or progesterone receptor (PR) expression, TNBC lacks effective targeted therapies and treatment regimens. Patients with mTNBC have fewer available treatment options and exhibit worse survival than early-stage TNBC patients. Furthermore, TNBC is a highly heterogeneous disease, resulting in substantial differences in the tumorigenesis, treatment response, and disease progression among patients (*Burstein et al., 2015*; *Jiang et al., 2019*; *Perou, 2011*), which undoubtedly poses significant challenges in prognostic prediction of the mTNBC and efficacy assessment for already limited treatment options. Unfortunately, reliable and tailored biomarkers to predict the prognosis and monitor the treatment response of patients with TNBC are yet to be established.

It has been a long-standing clinical management model to predict the prognosis of patients with mTNBC and guide treatment decision-making through imaging examination and mutational features obtained by tumor biopsy. Radiological examination usually only provides the external characterization of the tumor but can not reveal the tumor internal molecular characteristics. Given the heterogeneity of TNBC, it is impossible to obtain an accurate and comprehensive picture of the mutational landscape using tissue biopsies unless repeated multiple biopsies (*Diaz and Bardelli, 2014*), and most patients are refractory to repetitive punctures.

Compared with tissue biopsies, 'liquid biopsies' collect and analyze tumor-derived substances, such as circulating tumor DNA (ctDNA), circulating tumor cells (CTCs), and exosomes (e.g. from the blood, cerebrospinal fluid, and urine) of cancer patients in a minimally invasive fashion (*Alix-Panabières and Pantel, 2016*; *Palmirotta et al., 2018*; *Poulet et al., 2019*). It can be used for early diagnosis of tumor patients, predicting tumor recurrence and metastasis, and evaluating the characteristics and clonal evolution of tumor genomes. ctDNA is a specific fraction of cell-free DNA (cfDNA) present in the plasma of apoptotic and necrotic tumor cells (*Swarup and Rajeswari, 2007*). Owing to particular biological origin and the potential for multiple repeat sampling, ctDNA is independent of tumor spatial and temporal heterogeneity, conveys more valuable information than a conventional tumor biopsy, and enables the dynamic monitoring of tumor burden and treatment response (*Campos-Carrillo et al., 2020*; *Chae and Oh, 2019*; *Dawson et al., 2013*; *Gerratana et al., 2021*). The value of ctDNA in accurately predicting drug resistance and clinical outcomes has also been noted (*Asante et al., 2020*; *Murtaza et al., 2013*).

Several studies have demonstrated the prognostic and predictive value of ctDNA for non-mTNBC during or after (neo)adjuvant therapy (*Cavallone et al., 2020*; *Kim et al., 2021*; *Lin et al., 2021*; *Ortolan et al., 2021*; *Riva et al., 2017*). Previously, researchers have also been relatively circumscribed, concentrated on evaluating specific copy number variants (CNVs) or ctDNA-based single mutation or clonal evolution or the ctDNA level to predict the prognosis of mTNBC patients and the efficacy of particular treatment regimens (*Barroso-Sousa et al., 2022*; *Chopra et al., 2020*; *Collier et al., 2021*; *Stover et al., 2018*; *Weber et al., 2021*; *Wongchenko et al., 2020*). Even a study showed that the ctDNA level had no prognostic impact on the survival of patients with mTNBC (*Madic et al., 2015*). A more comprehensive study of the mutational information and related markers embodied in ctDNA as well as a consensus on the predictive role of ctDNA in mTNBC are needed to apply ctDNA in clinical practice.

Hence, in this study, we investigated the mutational characteristics of ctDNA and ctDNA-related markers in mTNBC using targeted, capture-based, next-generation sequencing (NGS), which offers

rapid identification and high coverage from a small blood sample. We aimed to dynamically and more comprehensively evaluate the value of ctDNA in predicting the prognosis and monitoring the treatment response of patients with mTNBC.

## Methods

### Study design

This was a prospective, single-center, observational study. All patients were enrolled at Shandong Cancer Hospital and Institute between 2018 and 2021.

Inclusion criteria were defined as follows: (1) Age >18 years and <80 years; (2) Histologically and radiologically confirmed mTNBC; (3) Patients with mTNBC who had progressed after ≤2 lines of chemotherapy, including those who were first diagnosed with mTNBC and had not yet received any therapy; (4) Chemotherapy-based treatment was determined as the upcoming therapeutic regimen for patients; (5) At least one measurable lesion according to Response Evaluation Criteria in Solid Tumors (RECIST 1.1) criteria; (6) Return for treatment on time and able to comply with study-related procedures including collection of plasma samples and treatment-related information, and follow-up.

Exclusion criteria were defined as follows: (1) Patients who had previously been treated with more than two lines of chemotherapy in an advanced stage; (2) Treated with local therapy (e.g. radiotherapy); (3) Lost to follow-up; (4) Failed to be evaluated for efficacy during treatment; (5) Unable to obtain plasma samples at baseline.

### Efficacy evaluation and follow-up

All target lesions were measured Using computed tomography (CT) or magnetic resonance imaging (MRI) after every two treatment cycles. Efficacy was evaluated according to RECIST 1.1 criteria. The endpoint observed in this study was progression-free survival (PFS), defined as the time interval between patient enrollment and confirmation of disease progression using CT/MRI scans or death from any cause. After enrollment, baseline characteristics of patients and chemotherapy-based treatment options jointly determined by the clinician and patient were recorded. Peripheral blood samples, hematological data of tumor markers such as carcinoembryonic antigen (CEA), cancer antigen 125 (CA125), and cancer antigen 153 (CA153), as well as the number and size of measurable lesions on CT/MRI were collected at the time of patient enrollment and each radiological examination for efficacy evaluation until the patients reached the endpoint of the study. During the investigation, we liaised with the patients and patients' physicians when patients revisited the hospital every time.

### Sample collection and preparation

A 10 mL sample of peripheral blood was collected into an EDTA anticoagulant tube (STRECK Cell-Free DNA BCT) from patients at different time points (i.e. before treatment, during treatment [treatment cycle 3, day 1], and at progression). Within 2 hr of collection, blood samples were centrifuged at 1600×$g$ for 10 min at 4°C to obtain plasma, followed by secondary centrifugation at 16,000×$g$ for 10 min at 4°C to obtain peripheral blood cells. Plasma and peripheral blood cells were stored at −80°C until ctDNA and genomic DNA (gDNA) were extracted. Paraffin-embedded primary or metastatic tumor tissues were collected before treatment and stored at room temperature for later use.

### DNA extraction and targeted capture-based NGS

ctDNA was extracted from peripheral blood using the QIAamp Circulating Nucleic Acid Kit (Qiagen, Germany), while tumor DNA (tDNA) was extracted from the paraffin-embedded tumor tissues using the AllPrep DNA/RNA FFPE Kit (50) (Qiagen, Germany). Standard control gDNA was extracted from white blood cells using the DNeasy Kit (Qiagen, Germany) according to the manufacturer's instructions. The sequencing library was prepared from the ctDNA and tDNA samples using the KAPA DNA Library Preparation Kit (KAPA Biosystems, USA), while the gDNA sequencing library was constructed using the Illumina TruSeq DNA Library Preparation Kit (Illumina, USA). Library concentration was determined using real-time quantitative PCR and the KAPA Library Quantification Kit (KAPA Biosystems, USA). The library fragments were then size-selected using agarose gel electrophoresis. A targeted NGS panel of 457 genes (*Supplementary file 1*), known to be frequently mutated in tumors, was designed to capture the target DNA fragments. All of the individuals were sequenced at a depth of at

least 20,000 X. Sequencing libraries were loaded onto a NovaSeq 6000 platform (Illumina, USA) with a 150 bp read length in paired-end mode.

## Sequencing data analysis

Quality control of the raw sequencing data involved using FASTP to trim adapters and remove low-quality sequences (*Chen et al., 2018*). The clean reads were aligned against the Ensemble GRCh37/hg19 reference genome using BWA software (*Li and Durbin, 2009*). PCR duplications were processed using the Gencore tool (*Chen et al., 2019*), and generated uniquely mapped reads. The SAMtools suite was used to detect single-nucleotide variants (SNVs), insertions, and deletions (*Li et al., 2009*). CNVs were called using CONTRA (*Li et al., 2012*), with copy number > 3 as the threshold of copy number gain and <1 as the threshold of copy number loss. The maximal tumor somatic variant allelic frequency (max-VAF) describes the highest mutated frequency of ctDNA detected in the cfDNA (*Maron et al., 2019*). The ctDNA fraction (ctDNA%) was calculated based on the autosomal somatic allele fractions. The mutant allele fraction (MAF) and ctDNA% are related as follows: MAF = (ctDNA ×1)/([(1 − ctDNA)×2] + [ctDNA ×1]); thus, ctDNA = 2/(1/MAF +1) (*Vandekerkhove et al., 2017*). The mutant-allele tumor heterogeneity (MATH) score was calculated as the percentage ratio of the width (median absolute deviation [MAD] scaled by a constant factor so that the expected MAD of a sample from a normal distribution equals the standard deviation) to the center of the distribution of MAFs among the tumor-specific mutated loci; thus, MATH = 100 × MAD/median (*Mroz and Rocco, 2013*). Tumor mutational burden (TMB) was defined as the number of non-synonymous somatic mutations per megabase of genome examined (*Chalmers et al., 2017*).

## Statistical analysis

The longest diameter (mm) of the tumor was measured by examining the radiological images. The response evaluation was carried out according to the Response Evaluation Criteria in Solid Tumors (RECIST) guidelines (version 1.1) (*Eisenhauer et al., 2009*). The Kaplan–Meier method was used for the survival analyses; the comparison of the median PFS was performed using the log-rank test and hazard ratios (HRs) from the Cox proportional hazards model. The optimal cut-off values for ctDNA%, MATH score, and TMB were determined by the R package 'survminer'. Univariate Cox regression was carried out to analyze the mutations related to PFS; only genes mutated in >5% of the patients were included in the analysis. The correlation between variables was analyzed using the Spearman correlation test, and group comparisons were performed using the Wilcoxon rank-sum test. Statistical analysis and data visualization were conducted using R (version 4.0.1). The statistical significance was defined as bilateral p<0.05.

# Results

## Study cohort and sample information

A total of 126 patients with mTNBC were considered for screening, and 70 patients were eventually included in the study. Fifty-six patients were excluded, comprising 21 patients who had previously been treated with more than two lines of chemotherapy, 16 patients who were treated with combination local therapy (e.g. chemotherapy with radiotherapy), 11 patients who were lost to follow-up, and 8 patients who failed to be evaluated for efficacy. Finally, 139 plasma samples (baseline samples from 70 patients and dynamic samples from 38/70 patients) from 70 patients and paired tumor tissues from 13 patients were collected and sequenced (*Figure 1A and B*). Some patients did not have surgery, or were operated on in other hospitals, or had no lesion suitable for biopsy, or the tumor tissue was too small, resulting in 57 patients not obtaining tumor tissue for sequencing. The blood samples during treatment were not collected on time, or the quality control of samples was unqualified or did not reach the endpoint of the study, resulting in 32 patients only having baseline blood samples. The patient baseline characteristics are shown in *Table 1*. The median age of all patients was 46 (26–75) years. Overall, 82.9% of patients were diagnosed with invasive ductal carcinoma, and most developed visceral metastases at study entry. *Table 2* shows that all patients received chemotherapy-based treatment with the most common chemotherapy drugs, such as gemcitabine, taxane, or platinum. Eleven patients were also treated with immunotherapy. The objective response rate was 38.6%, and 4, 23, 31, and 12 patients had complete response (CR), partial response (PR), stable disease (SD),

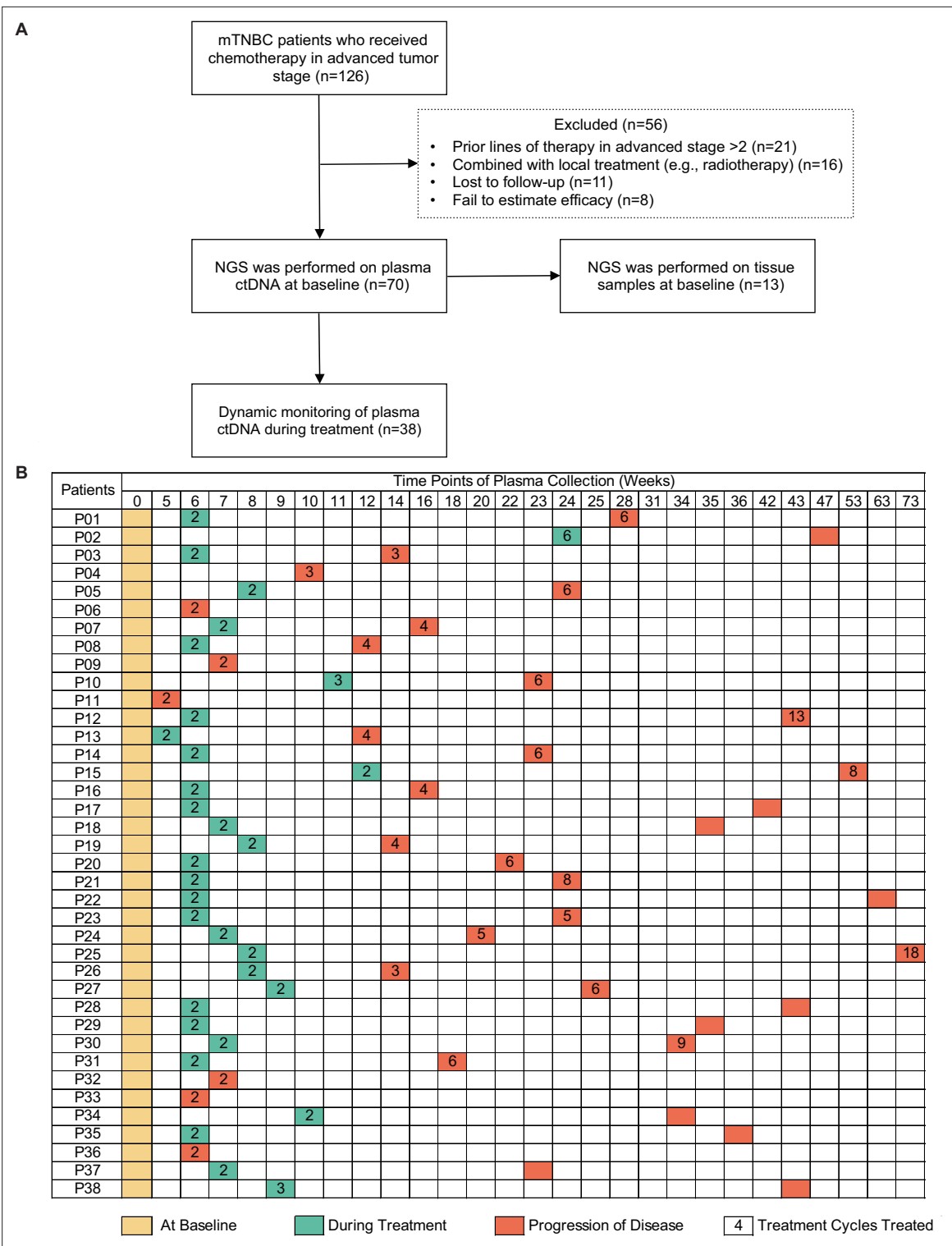

**Figure 1.** Study design and sample collection. (**A**) Study flowchart. After excluding 56 patients, 70 patients with mTNBC were included in the final analysis. Baseline blood samples were collected from all patients (n=70), and paired tumor tissues were collected from 13 patients for NGS. (**B**) Blood-sample-derived ctDNA was dynamically monitored at baseline (yellow), during treatment (green), and at progression (orange) for 38 of the 70 patients. ctDNA, circulating tumor DNA; mTNBC, metastatic triple-negative breast cancer; NGS, next-generation sequencing.

The online version of this article includes the following figure supplement(s) for figure 1:

**Figure supplement 1.** Kaplan-Meier analysis of PFS in total cohort and treatment-related subgroups of patients.

**Table 1.** The baseline characteristics in study population.

| Characteristics | All patients-no. (%) (n=70) | Dynamic monitoring patients-no. (%) (n=38) |
|---|---|---|
| Age (yrs), median (range) | 46 (26–75) | 47 (27–75) |
| ≤50 | 45 (64.3) | 26 (68.4) |
| >50 | 25 (35.7) | 12 (31.6) |
| Histopathologic diagnosis | | |
| Invasive ductal carcinoma | 58 (82.9) | 29 (76.3) |
| Other | 10 (14.3) | 7 (18.4) |
| NA | 2 (2.9) | 2 (5.3) |
| Pathological grade | | |
| I-II | 12 (17.1) | 10 (26.3) |
| III | 39 (55.7) | 20 (52.6) |
| NA | 19 (27.1) | 8 (21.1) |
| Disease stage at initial diagnosis | | |
| I | 7 (10.0) | 5 (13.2) |
| II | 20 (28.6) | 10 (26.3) |
| III | 33 (47.1) | 20 (52.6) |
| IV | 7 (10.0) | 3 (7.9) |
| NA | 3 (4.3) | 0 |
| Disease-free interval (months) | | |
| ≤12 (including stage IV at initial diagnosis) | 23 (32.9) | 11 (28.9) |
| >12 | 47 (67.1) | 27 (71.1) |
| Sites of metastasis | | |
| Visceral | 56 (80.0) | 33 (86.8) |
| Non-visceral | 14 (20.0) | 5 (13.2) |
| Previous lines of chemotherapy during metastatic stage | | |
| 0 | 46 (65.7) | 22 (57.9) |
| 1 | 19 (27.1) | 11 (28.9) |
| 2 | 5 (7.1) | 5 (13.2) |

and progressive disease (PD), respectively. The median PFS (mPFS) for all patients was 6.15 months (*Figure 1—figure supplement 1A*). There was no significant difference in PFS among different treatment lines or regimens (*Figure 1—figure supplement 1B-H*), although patients treated in the first line had a trend towards improved survival compared with those treated in the second and third lines.

## Mutation characteristics of patients with mTNBC

Plasma samples were obtained from 70 patients with mTNBC before treatment and submitted for targeted NGS. In total, 203 mutated genes were identified using our panel of 457 genes, including 301 missense mutations, 45 frame-shift indels, 16 in-frame indels, 13 splice-site mutations, and 31 stop-gain mutations. The ten most frequently mutated genes were *TP53* (69%), *PIK3CA* (24%), *ARID1A* (9%), *KMT2C* (9%), *CIC* (7%), *KMT2D* (7%), *NOTCH4* (7%), *PBRM1* (7%), *PTEN* (7%), and *DNMT3A* (6%). In addition, gene CNVs were detected in 351/457 genes, 296 of which showed copy number gain (CNG), 38 showed copy number loss, and 17 had both gain and loss mutations. The most

**Table 2.** The treatment characteristics and responses of patients.

| Treatment characteristics and responses | All patients-no. (%) (n=70) | Dynamic monitoring patients-no. (%) (n=38) |
|---|---|---|
| Treatment regimens | | |
| Contained gemcitabine +platinum | 17 (24.3) | 10 (26.3) |
| Contained taxane +platinum | 16 (22.9) | 7 (18.4) |
| Contained vinorelbine +platinum | 4 (5.7) | 1 (2.6) |
| Contained taxane but no platinum | 22 (31.4) | 14 (36.8) |
| Other | 11 (15.7) | 6 (15.8) |
| Treatment modalities | | |
| Immunotherapy +chemotherapy | 11 (15.7) | 6 (15.8) |
| Chemotherapy | 59 (84.3) | 32 (84.2) |
| Treatment responses | | |
| CR | 4 (5.7) | 4 (10.5) |
| PR | 23 (32.9) | 9 (23.7) |
| SD | 31 (44.3) | 18 (47.4) |
| PD | 12 (17.1) | 7 (18.4) |

prevalent genes with CNVs were *HLA-C* (50%), *HLA-A* (33%), *HLA-B* (33%), *HLA-DRB1* (23%), *SPPL3* (21%), *BTG2* (19%), *C8orf34* (16%), *HLA-DQA1* (16%), *HLA-E* (16%), and *LTBP1* (16%) (*Figure 2A*).

We also used NGS to evaluate the discrepancy and consistency of genomic alterations in ctDNA samples and paired tumor tissues from 13 patients. The mutation frequency in plasma ctDNA was significantly lower than that in the tumor tissues (0.049% ± 0.113% *vs* 0.168 ± 0.173%, p<0.001) (*Figure 2—figure supplement 1*). A total of 115 mutations in 85 genes were detected, which included 84 mutations in 63 genes from plasma ctDNA and 81 mutations in 55 genes from tDNA. The number of ctDNA-specific and tDNA-specific mutated genes was 37 in both cases. Hence, the concordance rate between mutations in ctDNA and tDNA was 98.75% (*Figure 2B*).

## A ctDNA+/− status correlates with the treatment response and survival of patients with mTNBC

Univariate Cox regression analysis showed that 12 mutated genes, including *HLA-B* (HR, 95% confidence interval [CI]: 1.89, 1.08–3.32), *BTG2* (HR, 95% CI: 2.23, 1.15–4.31), *MCL1* (HR, 95% CI: 2.31, 1.1–4.84), *H3F3A* (HR, 95% CI: 2.36, 1.08–5.16), *MYC* (HR, 95% CI: 3.45, 1.58–7.54), *KMT2C* (HR, 95% CI: 2.75, 1.14–6.63), *KYAT3* (HR, 95% CI: 3.66, 1.48–9.04), *ARID4B* (HR, 95% CI: 4.04, 1.51–10.76), *CD22* (HR, 95% CI: 3.66, 1.25–10.68), *TGFB1* (HR, 95% CI: 3.94, 1.35–11.56), *SGK1* (HR, 95% CI: 3.37, 1.15–9.82), and *RSPO2* (HR, 95% CI: 4.19, 1.42–12.34), indicated a higher risk for recurrence or progression in patients with mTNBC (*Figure 3A*). Moreover, these 12 mutated genes were significantly associated with worse survival (*Figure 3—figure supplement 1*).

ctDNA was collected and evaluated at different time points, and a plasma ctDNA sample with at least one of the above 12 prognosis-relevant mutated genes was defined as ctDNA-positive (ctDNA+) (*Figure 3B*). The right half of *Figure 3B* shows that the mPFS in 70 patients with mTNBC was 6.15 months at a median follow-up of 19.13 months. As shown, ctDNA+ patients tended to have a shorter survival duration and less clinical benefit. At baseline, the ctDNA+ rates were 25%, 43%, 55%, and 33% in patients with CR, PR, SD, and PD, respectively (*Figure 3B*). By comparing the ctDNA+/− status at all the time points, we found that the amount of ctDNA+ positively correlated with a worse treatment response. The proportion of ctDNA+ at different time points was 46% (at baseline), 29% (during treatment), and 44% (at progress) (*Figure 3B*), while that in the different treatment response groups was 0% (CR), 22% (PR), 39% (SD), and 45% (PD; *Figure 3C*). Before treatment, the tumor size of the ctDNA+ group was significantly larger than that of the ctDNA− group (52.56±34.65 mm *vs.*

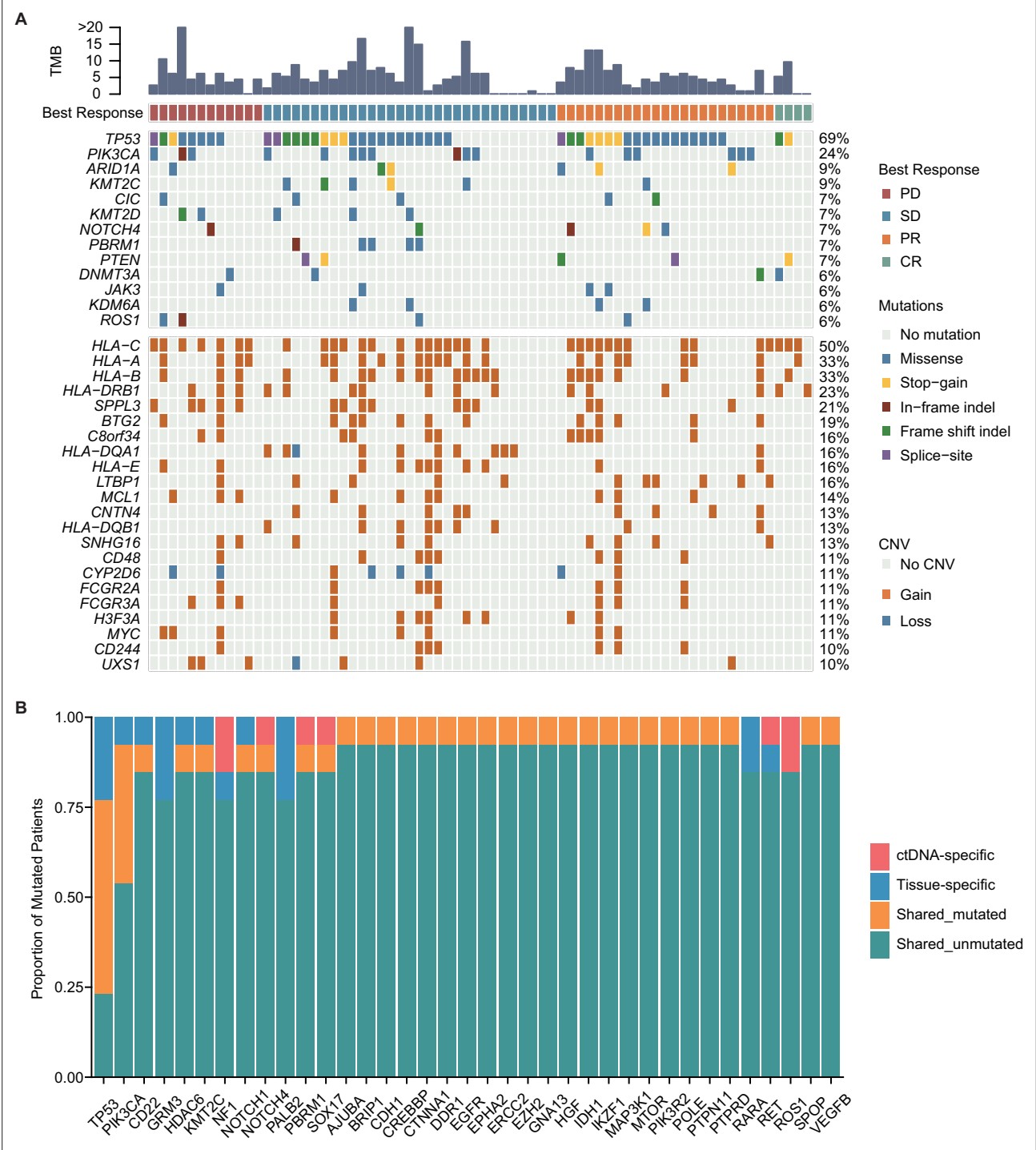

**Figure 2.** Mutation characteristics of patients with mTNBC. (**A**) The landscape of ctDNA mutations in 70 patients with mTNBC before treatment initiation. The patients (n=70) were divided into four groups (PD, SD, PR, and CR) according to the best treatment response (from left to right). The top half of the figure shows SNVs with a mutation frequency ≥5%, and the bottom half shows CNVs with a mutation frequency ≥10%; the different colored rectangles represent different types of variation. (**B**) Concordance between the genomic alterations in the blood-derived ctDNA and the tissue-derived tDNA. The mutated genes detected in at least two samples are shown here. The concordance rate = shared mutated genes/(all genes × the number of comparisons)×100% = (1 – [ctDNA-specific and tissue-specific mutated genes]/[all genes × the number of comparisons])×100% = (1 – [37+37]/[457×13])×100% = 98.75%. CNVs, copy number variants; CR, complete response; ctDNA, circulating tumor DNA; tDNA, tumor DNA; mTNBC, metastatic triple-negative breast cancer; PD, progressive disease; PR, partial response; SD, stable disease; SNVs, single-nucleotide variants.

The online version of this article includes the following figure supplement(s) for figure 2:

**Figure supplement 1.** Comparison of mutation frequency between ctDNA and tissue.

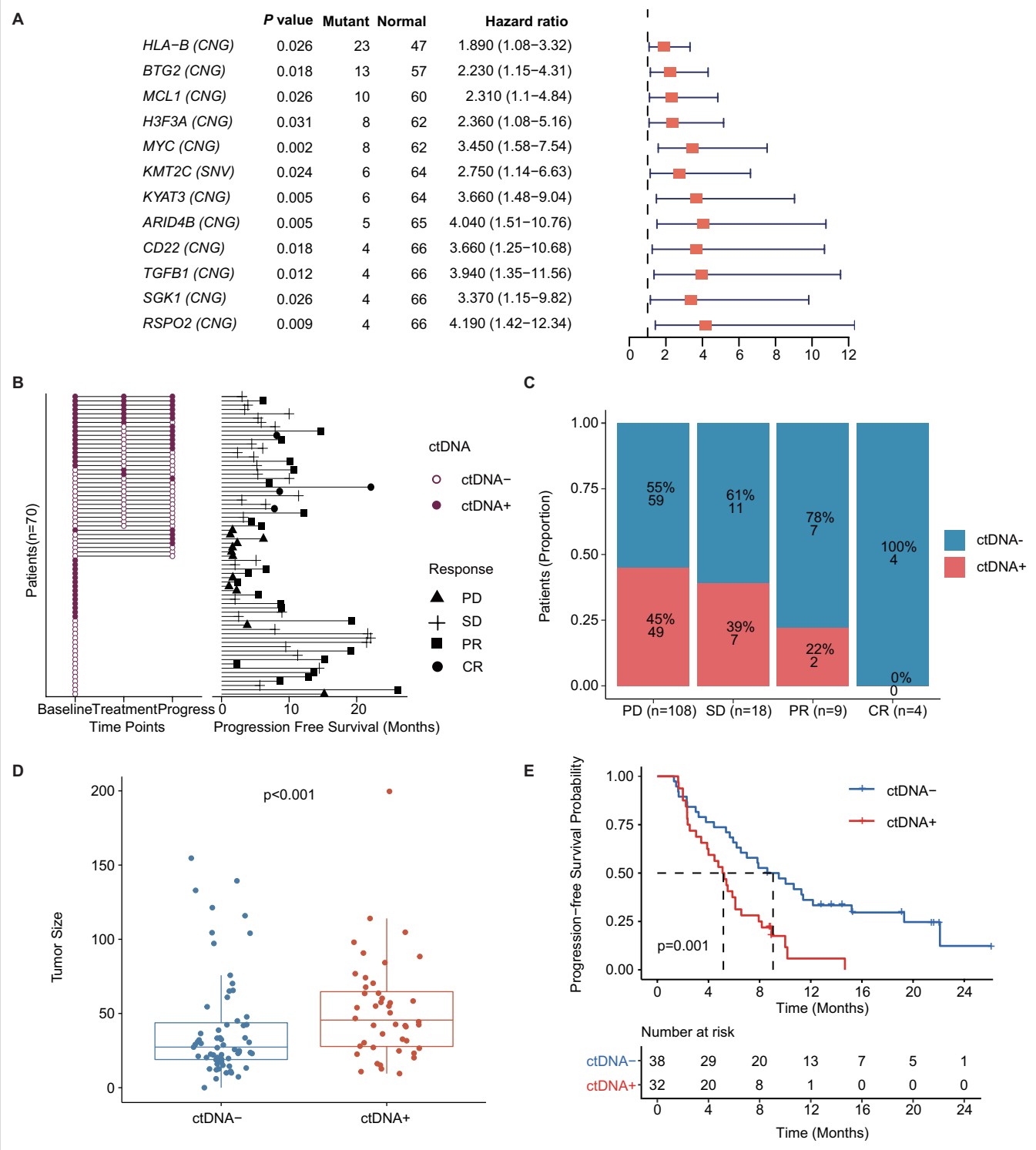

**Figure 3.** Prognostic relevance of mutations in patients with mTNBC. (**A**) Twelve mutated genes, comprising *HLA-B* (CNG), *BTG2* (CNG), *MCL1* (CNG), *H3F3A* (CNG), *MYC* (CNG), *KMT2C* (SNV), *KYAT3* (CNG), *ARID4B* (CNG), *CD22* (CNG), *TGFB1* (CNG), *SGK1* (CNG), and *RSPO2* (CNG) were identified as being associated with a higher risk of recurrence or progression in patients with mTNBC (all had HRs >1 and a p<0.05). (**B**) The left half of the figure summarizes the ctDNA status of all patients (n=70) at different time points; 38 patients also had their ctDNA status recorded during treatment and at progression. Solid dots represent ctDNA+ patients, while unfilled dots represent ctDNA– patients. The length of line segments in the right half of the figure denotes the PFS of patients, whereby the bars indicate the best response (PD, SD, PR, or CR) observed during treatment. (**C**) Comparison of ctDNA status (ctDNA+, red; ctDNA–, blue) among all the blood samples (n=139) from patients with different treatment responses (PD, SD, PR, or CR).

*Figure 3 continued on next page*

*Figure 3 continued*

(**D**) The tumor size of the ctDNA+ group at baseline was significantly greater than that of the ctDNA− group at baseline. (**E**) ctDNA+ at baseline was significantly associated with a shorter PFS. CNG, copy number gain; CR, complete response; ctDNA, circulating tumor DNA; ctDNA−, ctDNA negative; ctDNA+, ctDNA positive; HR, hazard ratio; mTNBC, metastatic triple-negative breast cancer; PD, progressive disease; PFS, progression-free survival; PR, partial response; SD, stable disease; SNV, single-nucleotide variant. A p-value <0.05 was used as a measure of statistical significance.

The online version of this article includes the following figure supplement(s) for figure 3:

**Figure supplement 1.** Kaplan–Meier analysis of the PFS of 12 prognosis-relevant mutated genes in patients with mTNBC.

40.18±35.49 mm, p<0.001; *Figure 3D*). We also found that patients who were ctDNA+ at baseline had a shorter PFS than those who were ctDNA− at baseline (5.16 months *vs.* 9.05 months, p=0.001; *Figure 3E*). Multivariate Cox regression analysis, which included multiple clinical factors and ctDNA status, showed that ctDNA+ was independently associated with a shorter PFS (HR, 95% CI: 2.67, 1.2–5.96; p=0.016; *Table 3*).

## Baseline ctDNA-related markers are associated with mTNBC patient survival and treatment response

To further explore the value of ctDNA in predicting clinical outcomes in mTNBC, we examined the association between the pre-treatment ctDNA-related markers (i.e. TMB, MATH score, and ctDNA%) and PFS and the treatment response. Thus, we performed Kaplan–Meier analyses of TMB, MATH score, ctDNA%, and PFS in patients with mTNBC. Although not statistically significant, TMB-high (≥2.63) patients tended to have a shorter mPFS than the TMB-low (<2.63) patients (5.87 months *vs.* 10.03 months, p=0.057; *Figure 4A*). Meanwhile, patients with a higher MATH score (≥6.316) had significantly shorter mPFS than patients with a lower MATH score (<6.316) (5.67 months *vs.* 11.27 months, p=0.007; *Figure 4B*). Moreover, the higher ctDNA% (≥0.05) patient group had a significantly shorter mPFS than the lower ctDNA% (<0.05) group (5.45 months *vs.* 12.17 months, p<0.001; *Figure 4C*). Patients with mTNBC were categorized into the PD, SD, PR, and CR groups according to their response to treatment. Further comparative analysis of baseline ctDNA parameters in different treatment response groups revealed that TMB was progressively lower across the four groups, showing a decreasing trend from the PD group to the CR group. Compared with other treatment

**Table 3.** Multivariate cox regression analysis of multiple clinical factors and ctDNA status with PFS of patients.

| Variable | HR (95 CI) | p value |
|---|---|---|
| Age (≤50 vs. > 50 years) | 1.21 (0.47–3.1) | 0.694 |
| Histopathologic diagnosis (IDC vs. non-IDC) | 0.64 (0.15–2.73) | 0.55 |
| Pathological grade (III vs. I-II) | 1.23 (0.53–2.86) | 0.629 |
| DFI (≤12 months vs. > 12 months) | 1.69 (0.6–4.71) | 0.319 |
| **ctDNA status (ctDNA+ vs. ctDNA−)** | **2.67 (1.2–5.96)** | **0.016** |
| T stage (T1 vs. T2 vs. T3) | 0.75 (0.38–1.47) | 0.397 |
| N stage (N1 vs. N2 vs. N3) | 0.79 (0.57–1.1) | 0.161 |
| CEA elevation (Yes vs. No) | 1.75 (0.64–4.81) | 0.278 |
| CA125 elevation (Yes vs. No) | 0.67 (0.24–1.88) | 0.445 |
| CA153 elevation (Yes vs. No) | 1.86 (0.62–5.57) | 0.268 |
| Ki-67 (≥30% vs.<30%) | 0.78 (0.33–1.88) | 0.582 |
| Site of metastasis (visceral vs. non-visceral) | 0.54 (0.18–1.61) | 0.267 |
| TMB (High vs. Low) | 1.08 (0.38–3.09) | 0.88 |

Footnotes: Disease-free interval (DFI) was defined as the time from the initial surgery to the disease progression. IDC: invasive ductal carcinoma.

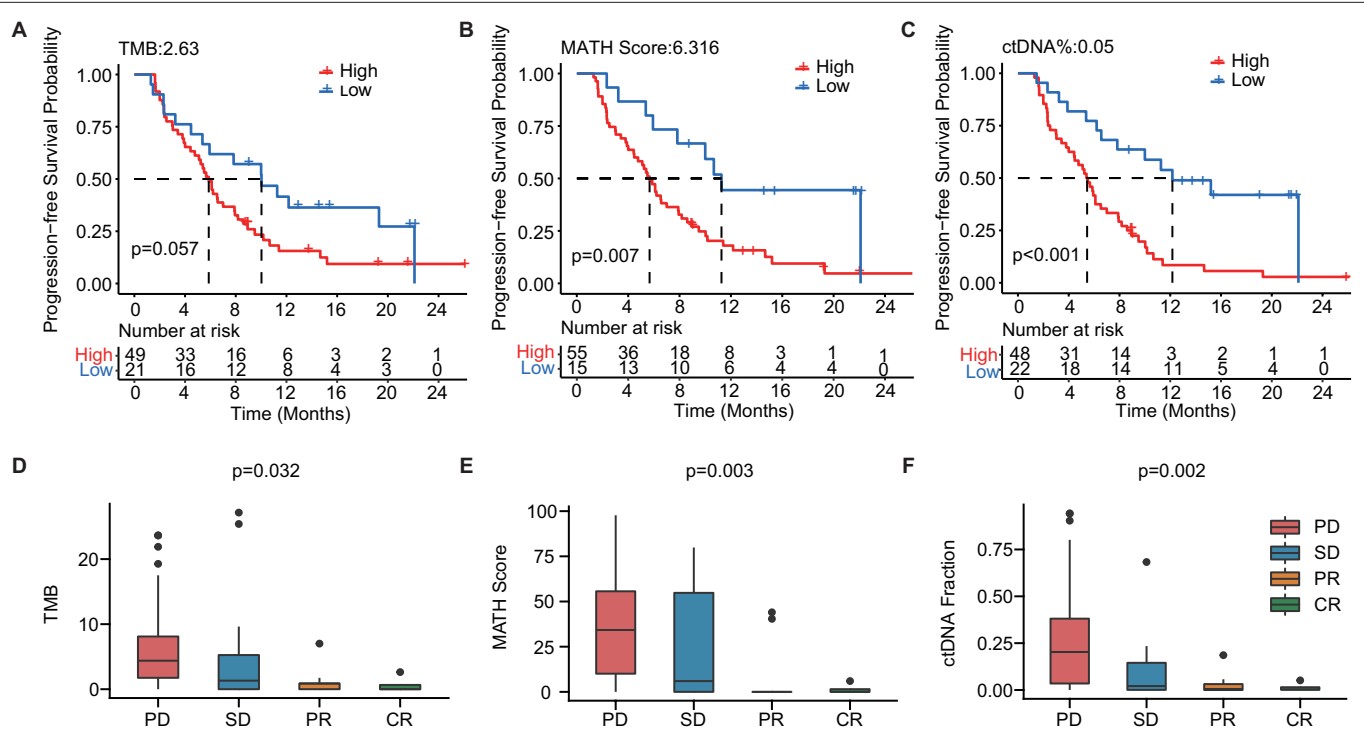

**Figure 4.** The baseline-ctDNA-derived TMB, MATH score, and ctDNA% were associated with the clinical outcomes of patients with mTNBC. Higher TMB (≥2.63) (**A**), MATH score (≥6.316) (**B**), and ctDNA% (≥0.05) (**C**) were linked to a shorter PFS. The optimal cut-off values for TMB, MATH score, and ctDNA% were determined using the R package 'survminer'. Comparison of TMB (**D**), MATH score (**E**), and ctDNA% (**F**) in patients with different treatment responses (PD, SD, PR, or CR). CR, complete response; ctDNA, circulating tumor DNA; ctDNA%, ctDNA fraction; mTNBC, metastatic triple-negative breast cancer; PFS, progression-free survival; PD, progressive disease; PR, partial response; SD, stable disease; TMB, tumor mutational burden. A p-value <0.05 was used as a measure of statistical significance.

response groups, the PD group had a larger TMB (p=0.032), greater MATH score (p=0.003), and higher ctDNA% (p=0.002; *Figure 4D–F*).

## Dynamic changes in ctDNA are associated with treatment response of patients with mTNBC

*Figure 5A–D* highlights the dynamic changes in ctDNA levels (i.e. mutations in 12 prognosis-relevant genes) and traditional tumor markers in each patient with PD (Patient 32), SD (Patient 31), PR (Patient 29), or CR (Patient 18). For instance, in Patient 32, the MAF of *MYC* ctDNA increased significantly and was accompanied by increased CA125 and CA153 levels but decreased CEA levels at the time of disease progression (*Figure 5A*). *Figure 5B–D* shows evidence of ctDNA mutations in Patient 31 (*BTG2, ARID4B, CD22, H3F3A, HLA-B, MCL1, MYC, RSPO2*), Patient 29 (*BTG2, ARID4B, H3F3A, HLA-B, MCL1, MYC*), and Patient 18 (*BTG2, ARID4B, H3F3A, HLA-B, SGK1*), their mutational rates dropped to the lowest level during the best response to treatment and rose again during progression. CA125 levels varied in line with treatment response and disease progression, but no similar fluctuations were observed in CA153 and CEA. Compared with these traditional tumor markers used in clinical for a long time, dynamic changes in ctDNA mutations seemed to mirror treatment-induced changes in tumor size better. Therefore, we analyzed the correlation between the levels of serum tumor markers and tumor size on CT/MRI scans during treatment (*Figure 5E*). We found that tumor size positively correlated with the MATH score (R=0.24, p=0.014) and ctDNA% (R=0.3, p=0.002) but not CEA, CA125, or CA153 levels. There were also strong positive correlations among the three ctDNA-related markers, TMB, MATH score, and ctDNA%. Moreover, the dynamic changes in ctDNA status may predict the prognosis of mTNBC. Kaplan–Meier analysis found that patients who remained ctDNA+ during dynamic monitoring had a shorter PFS than those who did not (3.90 months *vs.*

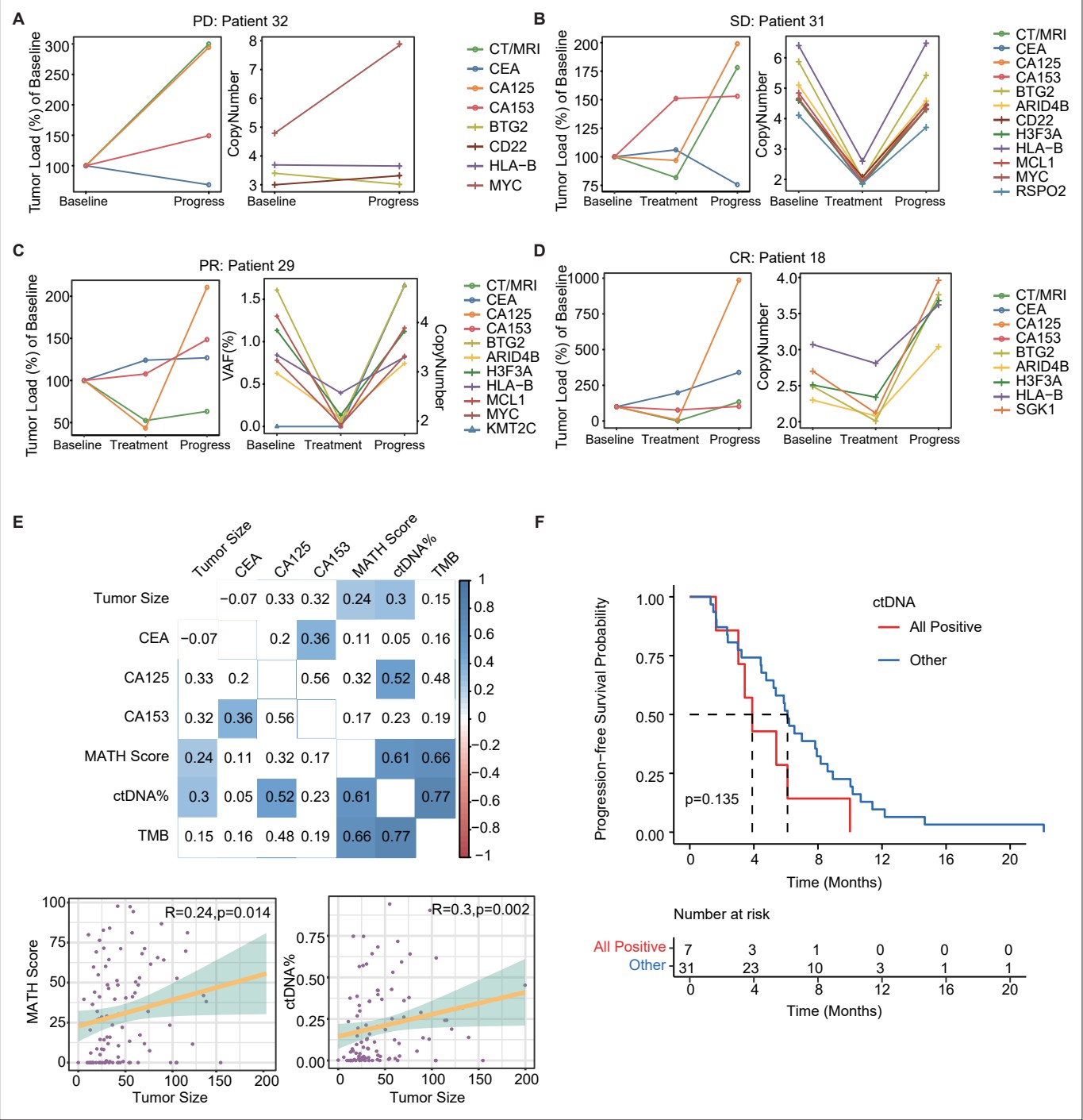

**Figure 5.** Dynamic ctDNA changes in patients with mTNBC. (**A–D**) Dynamic changes in tumor size were observed by radiological examination (CT/MRI) and the conventional tumor markers CEA, CA125, and CA153 (left side) or the VAF/copy number of 12 prognosis-relevant genes (right side) in four patients who each had a different best treatment response: PD (Patient 32), SD (Patient 31), PR (Patient 29), and CR (Patient 18). (**E**) The correlation between tumor size (measured using CT/MRI) and conventional tumor markers (CEA, CA125, CA153) or ctDNA parameters (TMB, MATH score, ctDNA%). Blue: positive correlation; red: negative correlation (the stronger the correlation, the darker the color). (**F**) Patients with a ctDNA+ status across all time points (All positive) tended to have a shorter PFS than those who were ctDNA− at least once during the dynamic monitoring process (Other). CR, complete response; ctDNA, circulating tumor DNA; ctDNA−, ctDNA negative; ctDNA+, ctDNA positive; ctDNA%, ctDNA fraction; mTNBC, metastatic triple-negative breast cancer; PFS, progression-free survival; PD, progressive disease; PR, partial response; SD, stable disease; TMB, tumor mutational burden; VAF, variant allelic frequency. A p-value <0.05 was used as a measure of statistical significance.

6.10 months, p=0.135; *Figure 5F*); however, this difference did not achieve statistical significance, most likely due to the limited sample size.

## Discussion

Since the advent of precision medicine, liquid biopsies have become more widely utilized in the clinical management of cancer. In recent years, ctDNA has become the focus of extensive research as a predictive biomarker of tumor progression and treatment response. Several studies have explored the promising applications of ctDNA in breast cancer. For example, some researchers have investigated the longitudinal dynamics of ctDNA in the treatment monitoring of metastatic breast cancer, while others have studied the prognostic and predictive value of ctDNA during neoadjuvant chemotherapy for TNBC (*Cavallone et al., 2020*; *Gerratana et al., 2021*; *Ortolan et al., 2021*; *Riva et al., 2017*). However, the application of ctDNA in monitoring mTNBC is rare in clinical practice.

In the current study, we performed targeted, capture-based NGS (with a 457-gene panel) on 139 plasma samples obtained by liquid biopsy from 70 patients with mTNBC. Thirteen paired tumor tissues were also analyzed to verify if ctDNA could be a feasible alternative to tumor-tissue-derived tDNA. This study demonstrated how a ctDNA-based platform could reliably reveal mutational profiles, stably predict the prognosis, and consistently monitor the treatment response of patients with mTNBC. This study also resolved the uncertainty of some current studies regarding the value of ctDNA and provided clear ctDNA-related predictive markers for mTNBC patients. By evaluating the mutational landscape of mTNBC in the Chinese population using ctDNA analysis, we showed that the most frequently mutated genes were *TP53*, *PIK3CA*, *ARID1A*, and *KMT2C*, and the most frequent CNV was detected in *HLA-C*, *HLA-A*, *HLA-B*, and *HLA-DRB1* (*Figure 2A*). These genes play significant roles in breast cancer tumorigenesis, progression, invasion, and metastasis. In the homogeneous population, the most common mutations detected in tumor tissue were *TP53*, followed by *PIK3CA*, *KMT2C*, and *PTEN*; however, the top-ranking CNVs were found in different genes to those identified in a previous report: *E2F3*, *IRS2*, *CCNE1*, *EGFR*, *NFIB*, *CCND1*, and *MYB* (*Jiang et al., 2019*). This difference may be due to the fact that the plasma-derived ctDNA contains smaller fragments than those that are typically found in tissue-derived tDNA. Meanwhile, compared with the results of other ctDNA identification studies, the distribution of frequent variants in the TNBC cohort was overall consistent (*Davis et al., 2020*; *Rong et al., 2020*; *Wang et al., 2021*). In the study by Chae et al., the concordance between all ctDNA- and tDNA-derived genes was also similar to that in our study (91.0%–94.2% *vs.* 98.75%; *Chae et al., 2017*; *Figure 2B*). The authors reported that the ctDNA-based assay had a high specificity, with a diagnostic accuracy of up to 80%. Although the mutation frequency of tDNA was higher than that of ctDNA in our study, the number of mutations detected in both types of DNA was similar. In addition, we detected 37 specific mutations in each of the ctDNA and tDNA groups, demonstrating the complementarity of blood-derived ctDNA and tissue-derived tDNA. Moreover, compared with tumor tissue analysis, ctDNA assays only require a small blood sample, can capture a variety of mutations (including SNVs and CNVs), and provide information on spatial tumor heterogeneity.

Using NGS, we identified 12 prognosis-relevant mutated genes associated with the shorter PFS of patients with mTNBC (*Figure 3—figure supplement 1*). Most of the 12 genes have been linked to breast cancer by previous studies. For instance, the aberrant expression of *KMT2C* (low expression) and *ARID4B* (high expression) contribute to the poor prognosis of patients with ER-positive breast cancer (*Sato and Akimoto, 2017*; *Zhang et al., 2021*). However, previous studies have shown that the upregulated expression of *BTG2* and *CD22* were associated with improved survival, which is not in agreement with our current findings (*Mascia et al., 2022*; *Zhang et al., 2013*); this may be due to the low number of patients included in this study. *TGFB1*, *SGK1*, *RSPO2,* and *MCL1* are implicated in the invasion, migration, growth, autophagy, and progression of TNBC, while *RSPO2* and *MCL1* overexpression is associated with shorter survival rates in patients with TNBC (*Coussy et al., 2017*; *Kim et al., 2015*; *Yang et al., 2014*; *Zhu et al., 2020*).

As a common driver of breast cancer, *MYC* amplification plays a role in emerging or acquired chemotherapy resistance during neoadjuvant treatment of TNBC. It can also synergize with *MCL1* amplification to maintain chemoresistance (*Lee et al., 2017*). *HLA-B*, a major histocompatibility complex (MHC) class I molecule, is involved in immunosurveillance against tumors, and its expression is correlated with the invasiveness and prognosis of breast cancer (*Concha et al., 1991*). To date, there have been no reports of an association between *H3F3A* or *KYAT3* and breast cancer. Previously,

*H3F3A* was identified as a driver gene in glioma, and its overexpression is linked to shorter survival rates and disease progression in lung cancer (*Felker and Broniscer, 2020*; *Park et al., 2016*). Thus, although the associations between some of the 12 mutated genes identified in our study and cancer are known, their roles in the prognosis of mTNBC need to be further defined.

We also explored the value of ctDNA status as a biomarker for predicting the prognosis and monitoring the treatment response of patients with mTNBC. We found that a ctDNA+ status was associated with a worse treatment response (*Figure 3B and C*). In addition, we showed that the ctDNA status at baseline could potentially discriminate between mTNBC patients with a high or low lesion load (*Figure 3D*). Moreover, ctDNA, as an independent prognostic factor, directly predicted the prognosis of patients without being influenced by other clinical factors of the patients (*Figure 3E*, *Table 3*). This indicates that the ctDNA status, associated with the presence of the 12 prognosis-relevant mutated genes, maybe a good guide to the prediction of clinical outcomes and the clinical management of TNBC. We further explored the optimal cut-off values for the ctDNA-based TMB, MATH score, and ctDNA% parameters at baseline using Kaplan–Meier analysis. A high MATH score (≥6.316) and high ctDNA% (≥0.05) were associated with a significantly shorter PFS (*Figure 4B and C*) and may therefore be related to the tumor burden of mTNBC. A study reported that most ctDNA fragments originate from metastases and not early-stage cancer, suggesting that ctDNA-based NGS may be more suitable for analyzing metastatic tumors (*Vandekerkhove et al., 2017*). Unlike ctDNA%, the MATH score represents tumor heterogeneity. A previous study found that TNBC was associated with a higher MATH score (*Ma et al., 2017*). Moreover, patients with higher MATH scores tend to have more diverse tumor cell clones and may be more prone to drug resistance and progression (*McDonald et al., 2019*; *Mroz and Rocco, 2013*). Thus, the MATH score could also potentially be used as a biomarker for mTNBC prognosis.

Breast cancer is a highly heterogeneous and dynamic disease; therefore, longitudinal monitoring and management are necessary (*Garcia-Murillas et al., 2015*). The predictive value of ctDNA has prompted further exploration of its feasibility in the dynamic monitoring of the efficacy of neoadjuvant therapy for breast cancer and in predicting the occurrence of distal metastasis and drug resistance (*Cavallone et al., 2020*; *Darrigues et al., 2021*; *Wang et al., 2021*). Here, we monitored ctDNA to track the dynamic changes in the 12 identified prognosis-related genes during the treatment of patients with mTNBC. The results showed that the elimination of these mutations or the reduction in the mutation rate of these genes was often associated with a better treatment response. Conversely, the reappearance of these mutations at a later time or an increase in their mutation rate signaled disease progression. Thus, we showed that ctDNA was sensitive and accurately reflected the treatment response and disease status of patients with mTNBC promptly. Conventional tumor markers have been widely used in clinical cancer management for a long time (*Anonymous, 1996*). We found that the serum CEA and CA153 levels contradicted the treatment response of Patient 31 at the mid-treatment time point (*Figure 5B*). The elevation of conventional tumor markers could indicate tumor changes and be interpreted as an early warning of disease progression or pseudo-progression. However, this pseudo-progression or 'tumor marker spike' is caused by extensive neoplastic cell necrosis induced by anti-tumor therapy. This phenomenon is observed in up to 30% of patients who respond to treatment (*Seregni et al., 2004*). NGS-mediated ctDNA detection identifies hundreds or thousands of mutations. Even if individual mutations were the result of a 'ctDNA spike' (similar to a 'tumor marker spike'), other mutations could be relied upon to mirror treatment response accurately. Moreover, we observed positive correlations between treatment response, MATH score, and ctDNA%, but not the CEA, CA125, and CA153 levels (*Figure 5E*). Hence, compared with conventional tumor markers, ctDNA dynamics may better reflect treatment response and progression in mTNBC.

Several limitations exist in our study. First, this was a single-center study with a small sample size. Second, the relatively short median follow-up duration was insufficient for capturing a clinically significant association between ctDNA mutations and overall survival. Third, several patients were lost to follow-up, which may have biased the results. Fourth, the potential influence of different treatment lines and regimens was not evaluated; nevertheless, no differences were found in survival based on these factors. Finally, compared with whole-exome sequencing or whole-genome sequencing, NGS with a panel of 457 selected genes, as used in our study, provided limited mutation data.

## Conclusions

ctDNA profiling is a good alternative to tumor tissue sequencing and provides valuable insights into the mutational landscape of mTNBC. Furthermore, our study addressed the value of ctDNA in predicting the prognosis and monitoring the treatment response of patients with mTNBC. The results revealed that higher ctDNA%, MATH score, TMB, ctDNA+ status, and mutation rate were associated with a poor prognosis and a worse treatment response in mTNBC. Moreover, the longitudinal monitoring of genetic biomarkers in ctDNA was more sensitive and accurate for discerning treatment response or progression than traditional tumor markers such as CEA, CA125, and CA153. Taken together, these findings will contribute to a better understanding of ctDNA in mTNBC and may facilitate the development of a more accurate and non-invasive clinical strategy for managing patients with this condition. However, larger clinical trials are necessary to validate our results.

## Acknowledgements

We sincerely thank the support of Yongsheng Wang, Pengfei Qiu, Binbin Cong, Peng Chen, Yanbing Liu, Chunjian Wang, Zhaopeng Zhang, Tong Zhao, Xiao Sun, Zhiyong Yu, Zhijun Huo, Xinzhao Wang, Shubin Song, Liang Zhang, Zhaoyun Liu, Fukai Wang, Chao Li, Xiang Song, Wenshu Zuo, Hui Fu, Meizhu Zheng, Ben Yang, Chao Han, Qian Shao, Xijun Liu, Jinzhi Wang, Wei Wang, Fengxiang Li, Yun Zhao, Linlin Wang, Bingjie Fan, Bing Zou, Zhenhua Gao, Xiangjiao Meng, Liyang Jiang, Zhengqiang Yang and Peng Xie. We also thank Liwen Bianji (Edanz) (https://www.liwenbianji.cn/) for editing a draft of this manuscript.

## Additional information

### Competing interests

Mu Su, Bo Yu, Jinxing Zhou: Affiliated with Berry Oncology Corporation. No financial interests to declare. The other authors declare that no competing interests exist.

### Funding

| Funder | Grant reference number | Author |
|---|---|---|
| National Natural Science Foundation of China | 81902713 | Huihui Li |
| Natural Science Foundation of Shandong Province | ZR2019LZL018 | Jinming Yu |
| Breast Disease Research Fund of Shandong Provincial Medical Association | YXH2020ZX066 | Huihui Li |
| Start-up Fund of Shandong Cancer Hospital | 2020-PYB10 | Huihui Li |
| Beijing Science and Technology Innovation Fund | KC2021-ZZ-0010-1 | Huihui Li |

The funders had no role in study design, data collection and interpretation, or the decision to submit the work for publication.

### Author contributions

Yajing Chi, Conceptualization, Formal analysis, Supervision, Investigation, Methodology, Writing – original draft, Project administration, Writing – review and editing; Mu Su, Formal analysis, Validation, Investigation, Visualization, Methodology, Writing – original draft; Dongdong Zhou, Resources, Data curation, Formal analysis, Validation, Visualization, Methodology; Fangchao Zheng, Baoxuan Zhang, Ling Qiang, Guohua Ren, Lihua Song, Bing Bu, Shu Fang, Resources, Data curation; Bo Yu, Resources, Data curation, Formal analysis, Investigation; Jinxing Zhou, Formal analysis, Investigation; Jinming Yu,

Conceptualization, Formal analysis, Supervision, Funding acquisition, Investigation, Project administration; Huihui Li, Conceptualization, Supervision, Funding acquisition, Project administration, Writing – review and editing

## Author ORCIDs
Huihui Li http://orcid.org/0000-0001-6040-3424

## Ethics
The study was approved by the Ethics Committee of Shandong Cancer Hospital and Institute (approval number: SDTHEC201806003) that collection of information, tumor tissues and blood samples within the ethical limitition of the patients, and conducted according to the Declaration of Helsinki. Written informed consent was obtained from all patients.

## Decision letter and Author response
Decision letter https://doi.org/10.7554/eLife.90198.sa1
Author response https://doi.org/10.7554/eLife.90198.sa2

---

# Additional files

## Supplementary files
• Supplementary file 1. The list of 457 genes detected in this study. Footnotes: Listed in this file are 457 genes, known to be frequently mutated in tumors, and were designed to compose a targeted NGS panel in this study to capture targeted DNA segments.

• MDAR checklist

## Data availability
The raw sequence data generated during the current study have been deposited in the China National Genomics Data Center-Genome Sequence Archive (GSA) (https://ngdc.cncb.ac.cn/gsa-human/) and are available with the accession code HRA002598.

The following dataset was generated:

| Author(s) | Year | Dataset title | Dataset URL | Database and Identifier |
|---|---|---|---|---|
| Li H | 2022 | HRA002598 | https://ngdc.cncb.ac.cn/gsa-human/browse/HRA002598 | China National Genomics Data Center, HRA002598 |

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
