## [Editor Report]

This valuable study advances our understanding of the predictive role of circulating tumor DNA (ctDNA) in the prognosis of patients with mTNBC as well as other malignant tumors. The evidence supporting the conclusions is solid, with rigorous analysis of the association between ctDNA (ctDNA-positive or not) with the progression-free survival (PFS) of patients. The work will be of broad interest to clinicians, medical researchers, and scientists working in breast cancer and cancer detection.

---

## [Decision Letter]

**Decision letter after peer review:**

Thank you for submitting your article "Dynamic Analysis of Circulating Tumor DNA to Predict the Prognosis and Monitor the Treatment Response of Patients with Metastatic Triple-negative Breast Cancer: a prospective study" for consideration by *eLife*. The following individual involved in the review of your submission has agreed to reveal their identity: Bo Tu (Reviewer #2).

Essential revisions:

1) Please elaborate the clinical usefulness of these ctDNA markers.

2) Please elaborate the post-treatment follow-up modality.

3) Please provide more details in the methods section for the prospective study. For instance, "Which center were these patients from? What are the inclusion and exclusion criteria? ".

*Reviewer #1 (Recommendations for the authors):*

1. ctDNA is considered a promising biomarker for early diagnosis. Does ctDNA reflect recurrence prior to CT scan? Longitudinal ctDNA analyses at multiple timepoints after treatment, and a comparison between ctDNA and CT scan results would potentially help to answer this question.

2. Constructing a model based on ctDNA-related markers and dynamic ctDNA levels, and using test sets to predict prognosis and treatment response would be an interesting idea to further support the clinical utility of ctDNA.

3. In the study cohort, patients who were treated with combination therapy (e.g. chemotherapy with radiotherapy) were excluded. However, eleven patients who were treated with both chemotherapy and immunotherapy were not excluded. Please detail the exclusion criteria.

4. Page 12, Line 280: "…while that in the different treatment response groups was 45% (PD), 39% (SD), 22% (PR), and 0% (CR)". The data was acquired during treatment or at progress? It seems that 55% of patients with PD do not have detectable ctDNA. Is it because of the low sensitivity of the detection method? This needs to be discussed.

5. The relationship between ctDNA with recurrence is not well addressed. How many patients relapsed? The recurrence rate among ctDNA^+^ patients and the ctDNA^+^ rate among patients with recurrence should be presented. A figure or a table indicating ctDNA and recurrence/progression status of each patient would be helpful.

6. Figure 5A-D shows the dynamic ctDNA changes of only four selected patients. It is not sufficient to support that the dynamic changes in ctDNA mutation better mirror treatment-induced changes in tumor size than traditional tumor markers. The results from other patients are needed.

7. Furthermore, is there any difference between the proportion of relapsed patients who detected all positive ctDNA during the dynamic monitoring with those that detected one positive or two positive?

8. How clinically useful these markers are is not adequately addressed. As only the original cohort is used, an external and independent patient cohort is needed to further validate the predictive value of ctDNA in mTNBC.

*Reviewer #2 (Recommendations for the authors):*

Several conclusions in the manuscript are not supported by the data and/or represent conceptual leaps, which must be addressed prior to publication. The specific questions and suggestions are as follows:

1. In the methods section, under patients and specimens, the post-treatment follow-up modality was not clearly described. How was tumor recurrence suspected? How were the patients followed-up?

2. In the manuscript, a sample with at least one of the twelve prognosis-relevant mutated genes was defined as ctDNA-positive (ctDNA+). The authors indicated that patients who were ctDNA^+^ at baseline had the larger tumor size, shorter PFS and less clinical benefit compared to those who were ctDNA− at baseline. I am wondering whether these clinical effects also depend on the amount of ctDNA+.

3. Nowadays, immunotherapy is playing an increasingly important role in breast cancer. I have noticed that there were some patients using immunotherapy in the cohort of this study. Have you considered focusing on the prediction of the efficacy of immunotherapy in these patients with ctDNA?

4. Predicting the prognosis and treatment response of these patients with mTNBC based on the ctDNA has been applied and demonstrated in this study. The authors also combed and recorded some clinical characteristics of the patients. Did the authors consider combining ctDNA-related markers with clinical characteristics to predict the prognosis and curative effect of the patients?

5. The total number of patients cohort in the study is 70, with baseline blood samples collected. However, only 38 patients were monitored dynamically. Moreover, 13/70 patients were sequenced with tissue samples. Could you please explain the reasons clearly in your manuscript.

6. In the manuscript, the proportion of ctDNA^+^ at different time points is different. In Figure 5, how to define the dynamic changes in ctDNA status (i.e., mutations in 12 prognosis-relevant genes)? What is the tendency of dynamic changes in ctDNA status in CPD, SD, PR, and CR groups?

7. Table 3, in multivariate cox regression analysis of multiple clinical factors and ctDNA status with PFS of patients, it seems that only variable factor ctDNA status is significantly correlated with PFS in patients. Could you please discuss more details in this part of result.

*Reviewer #3 (Recommendations for the authors):*

Specific comments

1. Figure 2 shows the landscape of ctDNA mutations in 70 patients with mTNBC prior to treatment initiation. Can these ctDNA mutation profiles recapitulate the mutation profiles of primary or metastatic tumors?

2. They found that ctDNA^+^ patients tended to have a shorter survival duration and less clinical benefit. For each patient, when the tumor progressed, did they find an increase of ctDNA positive rates?

3. More details should be included in the Methods section. For example, they stated that "Between 2018 and 2021, patients with mTNBC who had progressed after {less than or equal to} 2 lines of chemotherapy were prospectively enrolled.". Which center were these patients from? What are the inclusion and exclusion criteria?

4. The title of this study is "Dynamic Analysis of Circulating Tumor DNA to Predict the Prognosis and Monitor the Treatment Response of Patients with Metastatic Triple-negative Breast Cancer: a prospective study". Therefore, all patients are stage IV. Why did they include "Disease stage at initial diagnosis" in Table 1?

5. What is the correlation between "ctDNA%" and "ctDNA+"?

6. More details should be added to the figure legends.

---

## [Author Response]

Essential revisions:1) Please elaborate the clinical usefulness of these ctDNA markers.

We appreciate the comments by reviewers. As we stated in results and conclusions, ctDNA profiling is a good alternative to tumor tissue sequencing and provides valuable insights into the mutational landscape of mTNBC in clinical settings. Furthermore, our study addressed the value of ctDNA in predicting the prognosis and monitoring the treatment response of patients with mTNBC. The results revealed that higher ctDNA%, MATH score, TMB, ctDNA^+^ status, and mutation rate were associated with a poor prognosis and a worse treatment response in mTNBC. Moreover, the longitudinal monitoring of genetic biomarkers in ctDNA was more sensitive and accurate for discerning treatment response or progression than traditional tumor markers such as CEA, CA125, and CA153.

2) Please elaborate the post-treatment follow-up modality.

Thanks for your suggestion. We add the post-treatment follow-up modality in the “Methods”.

3) Please provide more details in the methods section for the prospective study. For instance, "Which center were these patients from? What are the inclusion and exclusion criteria? ".

Thanks for your suggestion. We really appreciate your suggestion. We add these details in the “Methods”.

Reviewer #1 (Recommendations for the authors):1. ctDNA is considered a promising biomarker for early diagnosis. Does ctDNA reflect recurrence prior to CT scan? Longitudinal ctDNA analyses at multiple timepoints after treatment, and a comparison between ctDNA and CT scan results would potentially help to answer this question.

Thank you for your valuable comment. The detection of ctDNA in the absence of metastasis or recurrent disease indicates molecular evidence of cancer, and have been defined as molecular residual disease (MRD). Several previous studies have reported that plasma ctDNA was detected ahead of clinical or radiologic relapse with a lead time of months or even years in breast cancer patients^1-3^. On the one hand, the PFS of mTNBC tend to be short, and on the other hand, intensive and regular longitudinal ctDNA analyses after treatment were not carried out in this study (only before treatment, during treatment [treatment cycle 3, day 1], and at progression), so it is not possible to determine the specific time of ctDNA before overt recurrence. But our findings suggest that ctDNA^+^ patients at baseline tend to have larger tumor sizes and poorer outcomes compared to ctDNA- patients. Moreover, the correlation analysis between dynamic ctDNA and tumor size on radiological scans found that ctDNA-related markers were also positively correlated with larger tumor size.

2. Constructing a model based on ctDNA-related markers and dynamic ctDNA levels, and using test sets to predict prognosis and treatment response would be an interesting idea to further support the clinical utility of ctDNA.

We gratefully appreciate your comment. In our analysis, we also tried to establish a prognostic and therapeutic prediction model for ctDNA-related markers, unfortunately, the model was not established due to the small sample size. Therefore, in the future, we intend to include more patients with mTNBC to explore the possibility of ctDNA-based prediction model.

3. In the study cohort, patients who were treated with combination therapy (e.g. chemotherapy with radiotherapy) were excluded. However, eleven patients who were treated with both chemotherapy and immunotherapy were not excluded. Please detail the exclusion criteria.

Thank you to point out this question. Here we excluded the patients who were combined with local therapy (e.g., radiotherapy) for tumor. Immunotherapy is a systemic treatment modality and therefore not excluded. We are sorry for not clarifying it clearly here. We have added detailed exclusion criteria in the "Method section" and corrected the "Study cohort and sample information" section.

4. Page 12, Line 280: "…while that in the different treatment response groups was 45% (PD), 39% (SD), 22% (PR), and 0% (CR)". The data was acquired during treatment or at progress? It seems that 55% of patients with PD do not have detectable ctDNA. Is it because of the low sensitivity of the detection method? This needs to be discussed.

Thank you for your question. "…while that in the different treatment response groups was 45% (PD), 39% (SD), 22% (PR), and 0% (CR)". The data was acquired from all the blood samples (n = 139) from 70 patients with different treatment responses (PD, SD, PR, or CR). The treatment response was the best observed during the follow-up. In other similar studies, patients who had ctDNA detected in blood were labeled ctDNA+.But in our study, ctDNA+/− status were redefined: A plasma ctDNA sample with at least one of the 12 prognosis-relevant mutated genes (*HLA-B*, *BTG2*, *MCL1*, *H3F3A*, *MYC*, *KMT2C*, *KYAT3*, *ARID4B*, *CD22*, *TGFB1*, *SGK1*, and *RSPO2*) which screened from 457 tumor-related genes was defined as ctDNA-positive (ctDNA+). Therefore, 55% of patients with PD just were not detected mutations in these 12 prognosis-relevant mutated genes, not ctDNA.

5. The relationship between ctDNA with recurrence is not well addressed. How many patients relapsed? The recurrence rate among ctDNA^+^ patients and the ctDNA^+^ rate among patients with recurrence should be presented. A figure or a table indicating ctDNA and recurrence/progression status of each patient would be helpful.

Thank you for your comment. Disease recurrence means that the observational endpoint of the study was reached. The endpoint observed in this study was progression-free survival (PFS), defined as the time interval between patient enrollment and confirmation of disease progression using CT/MRI scans or death from any cause. It can be seen in Figure 3E, by the end of the study, 90.6% (29/32) of the patients who were ctDNA^+^ at baseline relapsed, while 73.6% (28/38) of the ctDNA− patients relapsed. The vertical bars on the survival curve denote 4 ctDNA^+^ patients and 10 ctDNA^+^ patients, respectively, had not relapsed until the end of the study.

6. Figure 5A-D shows the dynamic ctDNA changes of only four selected patients. It is not sufficient to support that the dynamic changes in ctDNA mutation better mirror treatment-induced changes in tumor size than traditional tumor markers. The results from other patients are needed.

Thank you for your comment. In Figure 5A-D, we only selected four patients as representatives of four different therapeutic groups, so that readers can intuitively and concretely see the dynamic changes of ctDNA. In order to sufficiently support that the dynamic changes in ctDNA mutation better mirror treatment-induced changes in tumor size than traditional tumor markers, we analyzed the correlation between the dynamic changes of ctDNA-related markers of all patients and the traditional tumor markers such as CEA and tumor size in Figure 5E. Furthermore, changes of ctDNA+/− status based 12 prognosis-relevant mutated genes and survival were analyzed for all patients in Figure 5F.

7. Furthermore, is there any difference between the proportion of relapsed patients who detected all positive ctDNA during the dynamic monitoring with those that detected one positive or two positive?

Thank you for your comment. We also analyzed the possibility you proposed in our analysis, but no statistically significant difference in survival was found. In Figure 5F, patients who remained ctDNA^+^ during dynamic monitoring had a shorter PFS than those who did not (3.90 months vs. 6.10 months, *P* = 0.135). However, this difference did not achieve statistical significance, most likely due to the limited sample size.

8. How clinically useful these markers are is not adequately addressed. As only the original cohort is used, an external and independent patient cohort is needed to further validate the predictive value of ctDNA in mTNBC.

Thank you for your comment. Based on the findings of this study, we can use ctDNA-related markers to predict the prognosis and therapeutic efficacy of mTNBC patients, with a sensitivity superior to that of traditional tumor markers (CEA, CA125, and CA153). Subsequently, it can provide more clinically valuable prediction for the treatment effect and prognosis of tumor patients. It provides a more effective and sensitive evaluation method for the treatment effect and prognosis prediction of tumor patients. In order to ensure the success rate of patient specimen collection and facilitate follow-up, we only conducted this study in our hospital. To control for timeliness bias, we included only patients with mTNBC from 2018 to 2021. Unfortunately, there were few patients who met the inclusion criteria in this study within three years, and there was not enough sample size to add an external cohort for verification. We have also noticed this issue. Therefore, it is very necessary to further expand the sample size from multiple centers to verify the results of this study, which will be our next step in preparing for the study.

Reviewer #2 (Recommendations for the authors):Several conclusions in the manuscript are not supported by the data and/or represent conceptual leaps, which must be addressed prior to publication. The specific questions and suggestions are as follows:1. In the methods section, under patients and specimens, the post-treatment follow-up modality was not clearly described. How was tumor recurrence suspected? How were the patients followed-up?

We gratefully appreciate your comment. Follow-up modality and confirmation of tumor recurrence were added in Methods section.

2. In the manuscript, a sample with at least one of the twelve prognosis-relevant mutated genes was defined as ctDNA-positive (ctDNA+). The authors indicated that patients who were ctDNA^+^ at baseline had the larger tumor size, shorter PFS and less clinical benefit compared to those who were ctDNA− at baseline. I am wondering whether these clinical effects also depend on the amount of ctDNA+.

Thank you for your comment. The amount of ctDNA^+^ was for the entire cohort, while the clinical effect is for a single patient. Therefore, we generally only focus on the relationship between ctDNA status (positive or not) and clinical effect, rather than the amount of ctDNA+. Further, as you assumed, the worse clinical effect, the higher the proportion of ctDNA^+^ patients in group (Figure 3C).

3. Nowadays, immunotherapy is playing an increasingly important role in breast cancer. I have noticed that there were some patients using immunotherapy in the cohort of this study. Have you considered focusing on the prediction of the efficacy of immunotherapy in these patients with ctDNA?

We gratefully appreciate your comment. As stated in the title, in this study, we focused on possibility of ctDNA for prediction of treatment efficacy and prognosis in patients with mTNBC receiving chemotherapy. In another study, we have explored the predictive value of ctDNA for the efficacy of immune checkpoint inhibitors in patients with mTNBC.

4. Predicting the prognosis and treatment response of these patients with mTNBC based on the ctDNA has been applied and demonstrated in this study. The authors also combed and recorded some clinical characteristics of the patients. Did the authors consider combining ctDNA-related markers with clinical characteristics to predict the prognosis and curative effect of the patients?

Thank you for your comment. You and another reviewer gave the same valuable advice. We also attempted to combine ctDNA-related markers with clinical characteristics to establish a prognostic and therapeutic prediction model. Unfortunately, due to small sample sizes the models were not established. In future studies, we will include more mTNBC patients and explore the possibility of predictive model based on ctDNA and clinical characteristics.

5. The total number of patients cohort in the study is 70, with baseline blood samples collected. However, only 38 patients were monitored dynamically. Moreover, 13/70 patients were sequenced with tissue samples. Could you please explain the reasons clearly in your manuscript.

Thank you point this question. In this study, 32 patients only collected baseline blood samples without collecting dynamic time point blood samples. The main reasons included: (1) Patients neglected to provide post-treatment blood samples on time; (2) The patient has not reached the end of the progression; (3) The blood samples collected were not qualified. Tissue samples were obtained from surgical specimens that trace the patient's primary lesion, which cannot be obtained since most of the patients did not undergo surgery in our hospital. In addition, some patients were first diagnosed with advanced breast cancer and have not undergone surgery, so no tumor tissue can be obtained. Moreover, some patients had no lesions suitable for biopsy, or the tumor tissues were too small for sequencing. In the end, we only obtained tumor tissue samples from 13 patients. We added the explanation in the manuscript.

6. In the manuscript, the proportion of ctDNA^+^ at different time points is different. In Figure 5, how to define the dynamic changes in ctDNA status (i.e., mutations in 12 prognosis-relevant genes)? What is the tendency of dynamic changes in ctDNA status in CPD, SD, PR, and CR groups?

Thank you for your comment. We presented the dynamic changes of 12 prognosis-relevant genes from 4 patients with different therapeutic effects in the graph on the right (Figure 5A-D). The lines with different colors represent detectable mutated genes, the vertical axis represents the copy number or mutation frequency of 12 prognosis-relevant genes, and the horizontal axis represents different time points. The dynamic changes of these 4 patients can respectively represent the characteristics of ctDNA dynamic changes in the PD, SD, PR, and CR groups, summarized as follows: the elimination of these mutations or the reduction in the mutation rate of these genes was often associated with a better treatment response. Conversely, reappearance of these mutations at a later time point or an increase in their mutation rate signaled disease progression.

7. Table 3, in multivariate cox regression analysis of multiple clinical factors and ctDNA status with PFS of patients, it seems that only variable factor ctDNA status is significantly correlated with PFS in patients. Could you please discuss more details in this part of result.

Thank you for your comment. We have discussed more details of this result.

Reviewer #3 (Recommendations for the authors):Specific comments1. Figure 2 shows the landscape of ctDNA mutations in 70 patients with mTNBC prior to treatment initiation. Can these ctDNA mutation profiles recapitulate the mutation profiles of primary or metastatic tumors?

Thank you for your comment. As we noted in the manuscript, "ctDNA is a specific fraction of cell-free DNA (cfDNA), which is present in the plasma of apoptotic and necrotic tumor cells. Owing to special biological origin and the potential for multiple repeat sampling, ctDNA is independent of tumor spatial and temporal heterogeneity, convey more valuable information than a conventional tumor biopsy and enable the dynamic monitoring of tumor burden and treatment response", therefore, ctDNA reflects a real-time genetic signature of the whole-body tumor including primary or metastatic tumors.

2. They found that ctDNA^+^ patients tended to have a shorter survival duration and less clinical benefit. For each patient, when the tumor progressed, did they find an increase of ctDNA positive rates?

Thank you for your comment. Our study found that the elimination of 12 prognosis-relevant genes mutations or the reduction in the mutation rate of these genes was often associated with a better treatment response. Conversely, reappearance of these mutations at a later time point or an increase in their mutation rate signaled disease progression not ctDNA positive rates. ctDNA positive rates in this study indicates the proportion of ctDNA^+^ patients among the whole cohort.

3. More details should be included in the Methods section. For example, they stated that "Between 2018 and 2021, patients with mTNBC who had progressed after {less than or equal to} 2 lines of chemotherapy were prospectively enrolled.". Which center were these patients from? What are the inclusion and exclusion criteria?

Thank you for your comment. All patients were enrolled at Shandong Cancer Hospital and Institute between 2018 and 2021.The inclusion and exclusion criteria have added in Methods section.

4. The title of this study is "Dynamic Analysis of Circulating Tumor DNA to Predict the Prognosis and Monitor the Treatment Response of Patients with Metastatic Triple-negative Breast Cancer: a prospective study". Therefore, all patients are stage IV. Why did they include "Disease stage at initial diagnosis" in Table 1?

Thank you for your comment. Although all the patients included in this study were mTNBC patients, some patients were initially diagnosed as non-metastatic triple negative breast cancer, just had progressed to advanced stage when they were enrolled, while some patients are diagnosed with mTNBC at the first time. We considered the impact of disease-free interval (DFI) on prognosis in multi-factor analysis, so it is necessary to check the disease stage at initial diagnosis of patients.

5. What is the correlation between "ctDNA%" and "ctDNA+"?

Thank you for your comment. The calculation of the ctDNA fraction (ctDNA%) was based on the autosomal somatic allele fractions and is defined as the proportion of ctDNA in cfDNA. ctDNA was collected and evaluated at different time points and a plasma ctDNA sample with at least one of the 12 prognosis-relevant mutated genes (*HLA-B*, *BTG2*, *MCL1*, *H3F3A*, *MYC*, *KMT2C*, *KYAT3*, *ARID4B*, *CD22*, *TGFB1*, *SGK1*, and *RSPO2*) which screened from 457 tumor-related genes was defined as ctDNA-positive (ctDNA+).Therefore, there is no direct relationship between ctDNA% and ctDNA+.

6. More details should be added to the figure legends.

Thank you for your comment. We have added more details in table and figure legends.

References

1. Coombes RC, Page K, Salari R, et al. Personalized Detection of Circulating Tumor DNA Antedates Breast Cancer Metastatic Recurrence. Clinical cancer research: an official journal of the American Association for Cancer Research. 2019;25(14):4255-4263.

2. Medford AJ, Moy B, Spring LM, Hurvitz SA, Turner NC, Bardia A. Molecular Residual Disease in Breast Cancer: Detection and Therapeutic Interception. Clinical cancer research: an official journal of the American Association for Cancer Research. 2023.

3. Nguyen Hoang VA, Nguyen ST, Nguyen TV, et al. Genetic landscape and personalized tracking of tumor mutations in Vietnamese women with breast cancer. Molecular oncology. 2023;17(4):598-610.